# Glacier Boundary Mapping Using Deep Learning Classification over Bara Shigri Glacier in Western Himalayas

**Vishakha Sood** [1,2], **Reet Kamal Tiwari** [2], **Sartajvir Singh** [2,3,*], **Ravneet Kaur** [4] **and Bikash Ranjan Parida** [5]

1 Aiotronics Automation, Palampur 176 061, Himachal Pradesh, India
2 Civil Engineering Department, Indian Institute of Technology (IIT), Ropar 140 001, Punjab, India
3 Chitkara University School of Engineering and Technology, Chitkara University, Baddi 174 103, Himachal Pradesh, India
4 APEX Institute of Technology, Department of Computer Science Engineering, Chandigarh University, Mohali 140 413, Punjab, India
5 Department of Geoinformatics, School of Natural Resource Management, Central University of Jharkhand, Ranchi 835 222, Jharkhand, India
* Correspondence: sartajvir.singh@chitkarauniversity.edu.in

**Abstract:** Glacier, snow, and ice are the essential components of the Himalayan cryosphere and provide a sustainable water source for different applications. Continuous and accurate monitoring of glaciers allows the forecasting analysis of natural hazards and water resource management. In past literature, different methodologies such as spectral unmixing, object-based detection, and a combination of various spectral indices are commonly utilized for mapping snow, ice, and glaciers. Most of these methods require human intervention in feature extraction, training of the models, and validation procedures, which may create bias in the implementation approaches. In this study, the deep learning classifier based on ENVINet5 (U-Net) architecture is demonstrated in the delineation of glacier boundaries along with snow/ice over the Bara Shigri glacier (Western Himalayas), Himachal Pradesh, India. Glacier monitoring with Landsat data takes the advantage of a long coverage period and finer spectral/spatial resolution with wide coverage on a larger scale. Moreover, deep learning utilizes the semantic segmentation network to extract glacier boundaries. Experimental outcomes confirm the effectiveness of deep learning (overall accuracy, 91.89% and Cohen's kappa coefficient, 0.8778) compared to the existing artificial neural network (ANN) model (overall accuracy, 88.38% and kappa coefficient, 0.8241) in generating accurate classified maps. This study is vital in the study of the cryosphere, hydrology, agriculture, climatology, and land-use/land-cover analysis.

**Keywords:** deep learning; ENVINet5 (U-Net); artificial neural network (ANN); cryosphere; glacier boundaries

## 1. Introduction

The Himalayan cryosphere is important for south Asian countries to provide the water supply for irrigation, hydropower generation, and domestic use. It also causes natural hazards such as snow avalanches, glacier lake outburst floods (GLOF), and landslides [1]. Therefore, the regular and accurate mapping of cryosphere components, i.e., snow cover, glaciers, ice sheets, permafrost, and ice caps, is vital for the management of natural resources and the prediction service [2]. Recent decades have witnessed the severe impact of climate change on glacier health (i.e., retreat or expansion of glacier boundary), which can be tracked by understanding the continuous status of the cryosphere [3,4]. Moreover, assessing the glacier boundaries is essential to performing accurate determinating of glacier lengh andpredictions of glacier-mediated disasters and climate change [5]. Unfortunately, detection of glacier boundaries over the Himalayas is challenging due to rugged topography and harsh climate conditions [6]. Remote sensing is the only feasible and cost-effective solution for the continuous monitoring of glaciers on a daily basis [7]. Among the various

remote sensing datasets, Landsat series data are commonly used for the repeated mapping of glaciers at a fine resolution [8].

In previous literature, various models have been developed to detect the changes in the glaciers such as the normalized-difference snow index (NDSI) [9], machine learning classifiers [10,11] and object-based classification [12–14] using remote sensing data. These methods, however, highly depend on the analyst's skill experience in feature extraction, training process, testing, and validation procedures [15]. Hence, current requirements involve automatic, self-learning, fast, accurate, and unbiased classification models. Some attempts have been made in the past to use deep learning models for different applications as it is one of the fastest-growing trends in big data analysis. Although deep learning is an extended and time-consuming process, it offers an autonomous and better outcome as compared to conventional machine learning models [16]. As a part of machine learning, deep learning involves hierarchical architectures with numerous hidden layers in the neural network (NN) as in deep NN (DNN). This architecture helps to extract a specific set of features [17], whereas a convolution neural network (CNN) extracts the low-level as well as high-level features [18]. It can be further improved by the utilization of top-layer output of the network to abstract the high-level semantic information, which is very challenging with machine learning methods [19].

Deep learning has already proven as a successful tool in solving the complex problems and found applications in surface classification [20], image recognition [21], semantic segmentation [22,23], object recognition [24], object detection [25], and despeckling [26]. Figure 1 represents a brief overview of selected deep learning models, e.g., convolution neural network (CNN), recurrent neural network (RNN), deep reinforcement learning (DRL), graph neural network (GNN), stacked autoencoder (SAE), and generative adversarial networks (GAN) [18,27,28]. Compared to machine learning, it offers numerous advantages such as the automated approach to training the network itself, less human intervention required, better results in complex situations, and no strict requirement to check the quality of data for training [19].

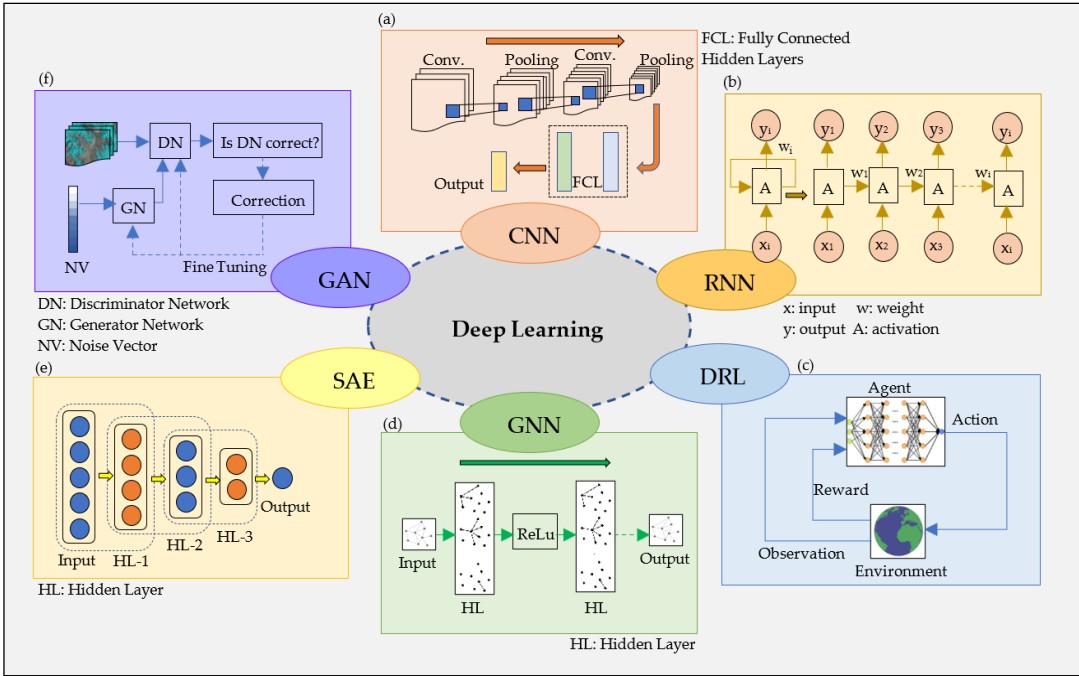

**Figure 1.** A brief overview of deep learning models: (**a**) convolution neural network (CNN), (**b**) recurrent neural network (RNN), (**c**) deep reinforcement learning (DRL), (**d**) graph neural network (GNN), (**e**) stacked autoencoder (SAE), and (**f**) generative adversarial networks (GAN).

With the availably of high-resolution satellite datasets and computations power, deep learning found numerous applications in earth observations through remote sensing such as data fusion, object detection, target detection, and scene classifications [29]. Recently, some attempts have been made to apply deep learning to glacier studies such as simulation and reconstruction of annual glacier-wide surface mass balance (SMB) using a deep learning-based artificial neural network (ANN) model [30], autonomous estimation of rock glaciers using a deep learning-based CNN model, object-based image analysis [31] and debris-covered glacier (DCG) mapping using a deep learning-based CNN model [32], alpine glacier mapping using a deep learning-based CNN model [33], and many more. Deep ANN as an alternative to the conventional model is the nonlinear model that solves the regression problems for predicting the various glacier components such as ice thickness [34] and mass balance [35]. The deep CNN model requires larger sample datasets to train the models and helps in the recognition of repeated patterns in different applications such as snow avalanches [36–39], permafrost thaw slumps [40], and sea-ice mapping [41]. However, very few studies have focused on glacier boundary mapping using deep learning models, which may be beneficial in climate change studies. Furthermore, deep learning is yet to be explored for different applications with different remote sensing datasets because the complexity increases with: (a) high spectral/spatial resolution to extract detailed information; (b) a large number of training samples to improve the recognition; and (c) depth of the deep learning model to learn complex distribution [19].

To exploit the power of deep learning to tackle these different challenges, the main focus of this study is to delineate the glacier boundaries along with snow and ice coverage using a deep learning classifier over a part of the western Himalayas. The objectives of the study are framed as (a) computation of snow/ice maps using ENVINet5 (based on U-Net) deep learning, (b) extraction of the glacier boundary, (c) comparative analysis with the existing ANN-based model, and (d) validation and accuracy assessment procedures for the classified outcomes. The U-net architecture-based ENVINet5 deep learning model has been utilized to detect the snow, ice, glaciers, and their boundaries using the Landsat-8 satellite dataset. This study was conducted over the Bara Shigri glacier to confirm the effectiveness of the ENVINet5-based deep learning model in the estimation of glacier boundaries.

## 2. Study Area and Data

This study was performed over a Bara Shigri glacier, located near the Chandra River basin, nearby Chandra Valley on the northern ridge of Pir-Panjal range, Lahaul-Spiti, Himachal Pradesh, India, using Landsat-8 satellite dataset as shown in Figure 2. This glacier is the second-longest glacier (28 km) in the Himalayas, after Gangotri, and situated between geographical coordinates 32°05′ N–32°17′ N latitude and 77°33′ N–77°48′ E longitude. It covers the 131 km$^2$ with a glacierized area ranging between 3920 m (13,071 ft) and 6550 m (20,876 ft) a.m.s.l. [42,43]. As the largest glacier in Himachal Pradesh (Indian State), it provides valuable information about climate change studies [44]. Various studies also explored the retreating of the Bara Shigri glacier with an average terminus retreat of 1100.2 ± 32.1 m (22.5 ± 0.7 m per year) from 1965 to 2014 [45,46].

The satellite dataset was acquired on a cloud-free clear daytime (20 September 2017) from Landsat-8 OLI (Operational Land Imager) with 30 m spatial resolution. The Shuttle Radar Topography Mission (SRTM)-based Digital Elevation Model (DEM, version 4) with 30 m spatial resolution data was utilized to analyze the topography of the Bara Shigri glacier as shown in Figure 3. Figure 3b,c represent the aspect (compass direction/azimuth, i.e., terrain surface faces, generally measured in degrees from north) and slope (steepness of terrain, i.e., degree of inclination of land surface to the horizontal plane), respectively, generated from STRM-DEM. This glacier has heterogenic surface characteristics, i.e., clean ice (located in the accumulation zone) and debris cover (located in the lower ablation zone).

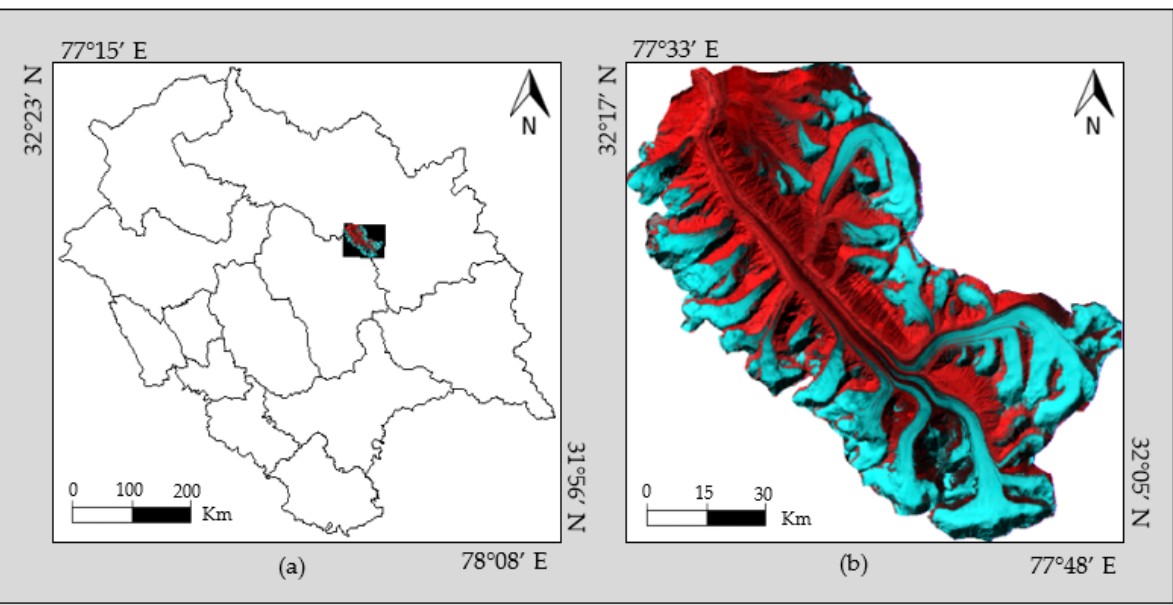

**Figure 2.** Location of the study area: (**a**) Himachal Pradesh, India, highlights the study area and (**b**) Landsat-8 imagery at RGB (621) band combination.

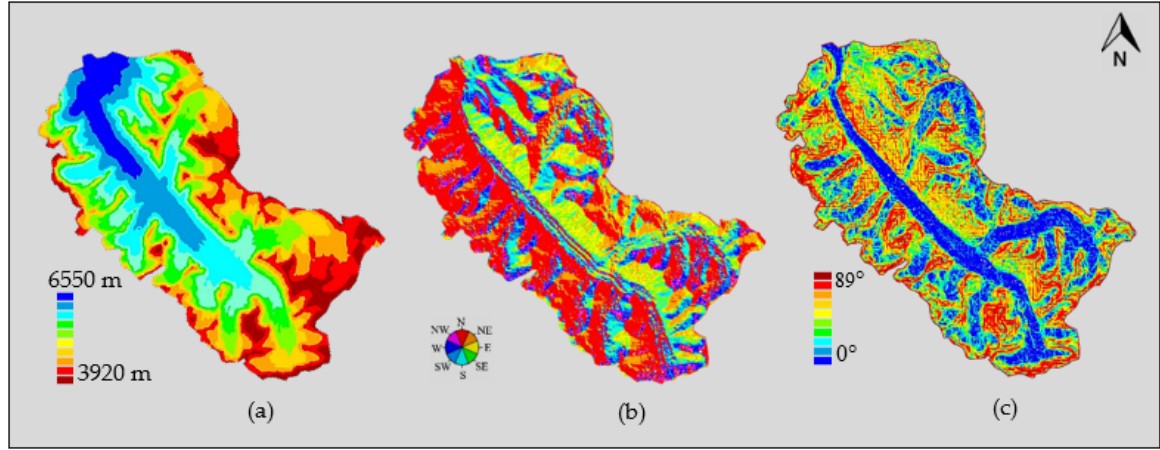

**Figure 3.** Representation of Bara Shigri glacier topography via (**a**) ASTER-based DEM; (**b**) aspect and (**c**) slope.

For validation purposes, very high-resolution data were acquired from the Pléiades constellation (Pléiades–HR 1A and Pléiades–HR 1B), Airbus Defense, and Space/Centre National d'Etudes Spatiales (CNES). Pléiades data were extracted from the Google Earth Engine (GEE) at a high spatial resolution of 0.5 m (PAN, panchromatic) and 2 m (MS, multispectral).

## 3. Methodology

The framework of the methodology comprises (a) preprocessing of the input dataset, i.e., multispectral Landsat-8; (b) implementation of the deep learning classifier that included training, testing, generation of activation or classified maps, and extraction of glacier boundaries; and (c) accuracy assessment of the classified maps generated from different deep learning classifiers. The detailed procedure of methodology is shown in Figure 4.

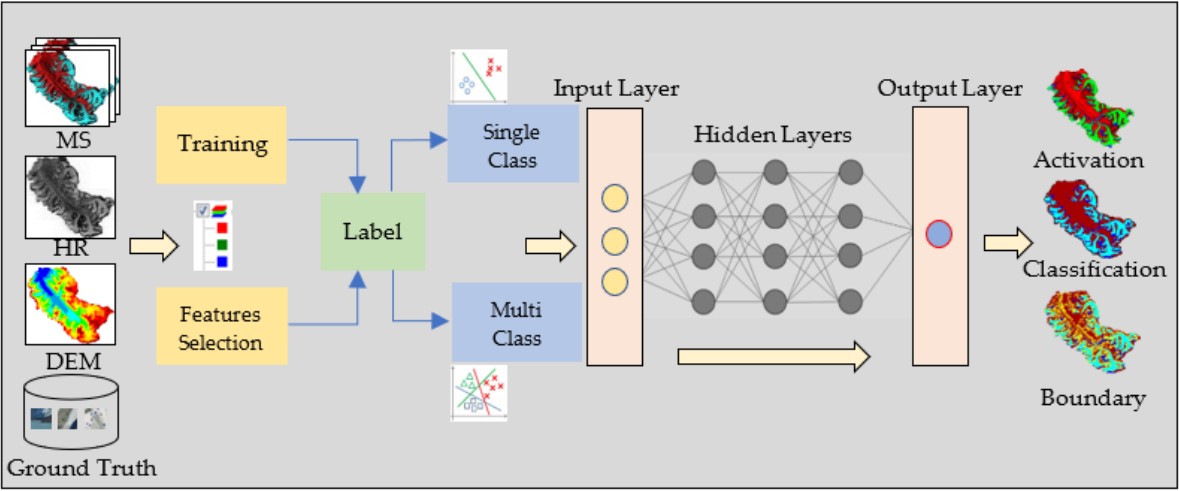

**Figure 4.** The flowchart of the deep learning model for remote sensing data classification.

### 3.1. Preprocessing of the Input Dataset

Initially, the Landsat-8 was preprocessed to eliminate the atmospheric/radiometric effects in both spatial and temporal terms, which exist due to the sensor's calibration or air temperature anomalies. The processing of the satellite includes two basic steps, i.e., (a) conversion of digital number (DN) values into surface radiance values and (b) conversion of radiance values into reflectance values. The DN to radiance $L$ conversion is achieved as follows [47–49]:

$$L = \frac{[(L_{max} - L_{min}) \times DN]}{1024} + L_{min} \tag{1}$$

where the terms $L_{max}$ and $L_{min}$ are maximum and minimum radiance, respectively. Afterward, the image-based atmospherically corrected reflectance $R$ is computed as follows [50,51]:

$$R = \frac{\pi \times L \times d^2}{E_s \times Cos\,\theta_s} \tag{2}$$

where $d$ and $E_s$ are Earth–sun distance and mean exoatmospheric solar irradiance, respectively. The $\theta_s$ represents the solar zenith angle.

### 3.2. Data Classification Using Deep Learning

Classification allows the quantitative analysis of remote sensing image data to categorize all the pixels in the imagery into a specific set of class categories. In previous literature, many machine learning or deep learning classifiers were developed and summarized by various authors in different applications with different remote sensing datasets [52,53]. Among the various classifiers, deep learning approaches attract the interest of many researchers due to their potential for efficient extraction of discriminative features and their representation in the real domain [54]. Some of the prominent deep learning-based classifiers are CNN, to learn the learning of low- and high-level features [54]; RNN, to solve the problem of gradient diminishing [55]; DRL, to learn the best actions for given states using a feedback–reward system [56]; GNN, to learn from unstructured data [57]; SAE, to automatically learn deep features [58]; and GAN, which contains the generator and a discriminator to learn a latent space [59].

Here, we have implemented the deep learning classifier based on ENVINet5 and ENVI Net-Multi. The deep learning version 1.1 is an add-on module in ENVI (Environment for Visualizing Images) image processing software version 5.6 (2020) image analysis. Figure 5 represents the ENVINet5 architecture: (a) input patch (gray color), (b) feature map (orange color); (c) 3 × 3 convolution (green color); (d) feature fusion (yellow color); (e) dimensionality reduction by maximum pooling (blue color); (f) co-convolution (red color); and (g) 1 × 1 convolution (purple color). This architecture is mask-based encoder–decoder

architecture to categorize the pixel data in imagery. In the ENVI deep learning model, the ENVINet5 and ENVI Net-Multi are specially designed for a single class and multiple classes, respectively. These architectures are based on the U-Net model (a channel-attention U-Net) with five levels with twenty-three convolutional layers [60]. The robustness of the model toward the distortion can be controlled by down-sampling. The restoration and decoding of the abstract features can be done by up-sampling.

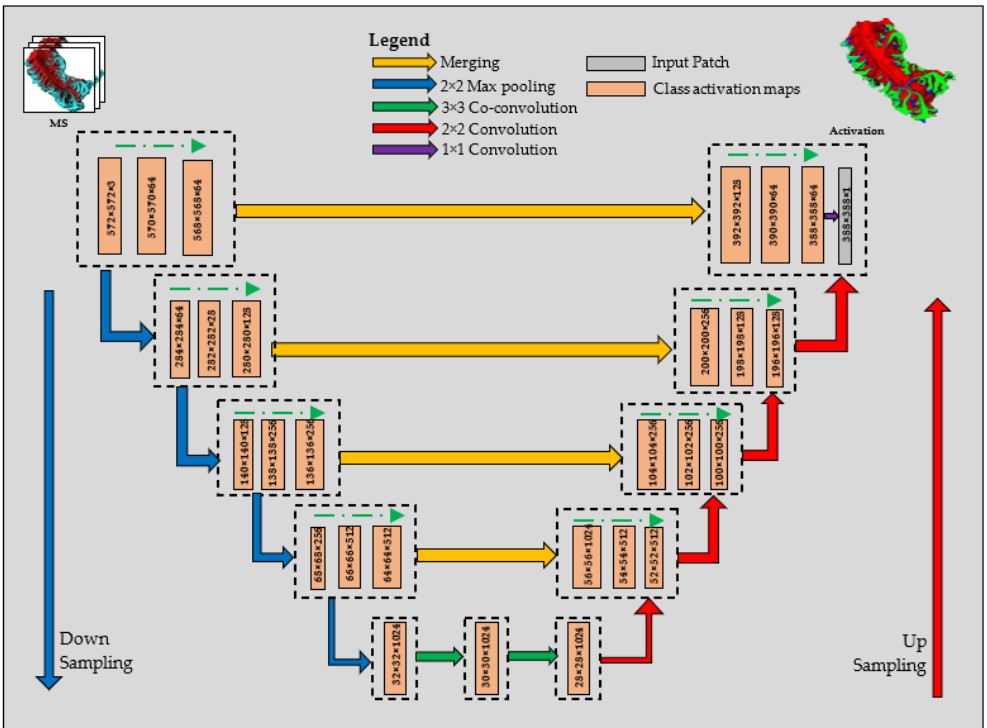

**Figure 5.** The architecture of Environment for Visualizing Images (ENVINet5), based on the U-Net.

Here, ENVINet-Multi was utilized to classify the multiple class categories. In this model, the multiple features, i.e., snow, ice, and barren (including the mixture of barren or trees), were extracted based on input dataset spectral and spatial properties with the knowledge of field data. To train the TensorFlow, we used multiple on-screen training samples (50–60 polygons) from each class category with respect to the knowledge of ground truth information. The various training parameters for ENVINet-Multi were set as (a) number of epochs: 25; (b) patch sampling rate: 16; (c) class weight: 2.5; (d) loss weight: 0.5; (e) number of patches per epoch: 200; (f) patch size: 464 × 464 pixels; and (g) number of patches per batch: 2.

In the training process, it repeatedly exposes label rasters to a model and thus converts the spectral and spatial information of the raster into the thematic map with the help of prepared training samples. The data are split into two parts, i.e., training (80%) and validation (20%) [61]. At last, the TensorFlow mask classification tool is utilized to classify the input dataset with the help of a trained model. This entire process may require a significant amount of time as per the computation availability. While implementing the ENVINet5-based deep learning on the Landsat-8 dataset, the computer specifications included the Intel Xeon 3.2 2400 MHz 8.25 4C CPU, 16 GB of RAM, 512 GB of SSD, and NVIDIA Quadro P620 2 GB (4) MDP GFX.

In the ENVINet5, the resolution of the image is down-sampled and up-sampled four times, as shown in Figure 5. The deep learning module v1.1 (available in ENVI v5.6) offers the advantages of a single trained model to classify the numerous inputs/images with respect same set of spectral and spatial properties. Deep learning leverages TensorFlow technology to train classification models. In addition to this, there is also an option to export the classes into a shape (boundary) file from thematic maps that can be used to

estimate a specific class category. The boundary files are also essential to distinguish the different classes. The deep learning module is designed to hide the complexity of CNN from image analysts.

### 3.3. Accuracy Assessment

To test the efficacy of ENVINet5-based deep learning, an accuracy assessment is computed for the classified dataset. The various important parameters of accuracy assessment involve the producer's accuracy (PA), i.e., probability of correct classification of reference pixels; user's accuracy (UA), i.e., probability of pixels falling under a precise class category; commission error (CE), i.e., classified sites probability for incorrect classification; omission error (OE), i.e., the probability to classify reference pixels correctly; overall accuracy (OA), i.e., collective accuracy map for all the classes; and Cohen's kappa coefficient (k), i.e., the distinction between actual and expected agreement, computed as follows:

$$k = \frac{N \sum_{i=1}^{r} m_{ii} - \sum_{i=1}^{r}(G_i C_i)}{N^2 - \sum_{i=1}^{r}(G_i C_i)} \tag{3}$$

where $N$ represents the total count of observations; $r$ shows the errors matrix count of rows and columns; $m_{ii}$ depicts the row and column observation belonging to class $i$; and $G_i$ represents the true values of class $i$, whereas $C_i$ shows the predicted values of class $i$. For sample selection, stratified random sampling is utilized to select the samples from a population, which can be partitioned into subpopulations. The outcome of the ENVINet5-based deep learning is compared with the conventional ANN model, i.e., feed-forward NN.

## 4. Results and Discussion

The ENVI v5.6 deep learning module v1.1 generates the activation rasters in the form of fractional maps or grayscale (value lies between '0' to '1') of each class category, i.e., snow, ice, and barren, as shown in Figure 6a–c. The value '0' means the minimum existence of the class category, and the value '1' means the maximum existence of the class category. These fractional maps allow the accurate detection of particular class categories. These fractional maps can be visualized in combined form by setting the ice in the blue plane, snow in the green plane, and barren in the red plane, as shown in Figure 6d. Furthermore, the class membership of each pixel can be measured by implementing the threshold algorithm. Here, we have used Otsu as an automatic threshold search method to generate the classified image from the activation rasters, as shown in Figure 6e. If the results are not satisfactory, the user can reset the threshold value or threshold algorithm (in the ENVI v5.6 deep learning module v1.1) to regenerate the accurate classified image from the activation raster instead of recomputing the entire deep learning classification process. Once the model is trained, it can also be used for different multitemporal images of the same satellite dataset with the same number of spectral bands only.

In addition, the ENVINet5 also allows for the creation of the polygon shapefile from the class activation raster in order to draw and estimate the boundaries around the different features, as shown in Figure 6f. ENVINet5 delivers a robust method for learning complex spatial and spectral information, which may help in extracting the essential features from a complex dataset, regardless of their different attributes, e.g., shape, color, size, and barren. For the cross-referencing, the outcomes of ENVINet5 were compared with the conventional ANN model, i.e., feed-forward NN, as shown in Figure 7. Figure 7a–c represent the ANN activation maps (i.e., snow, ice, and barren) computer-generated via the ANN model. Figure 7d–f represent the combined form (red plane: barren; green plane: snow; and blue plane: ice) of fractional maps, classified maps, and boundary maps.

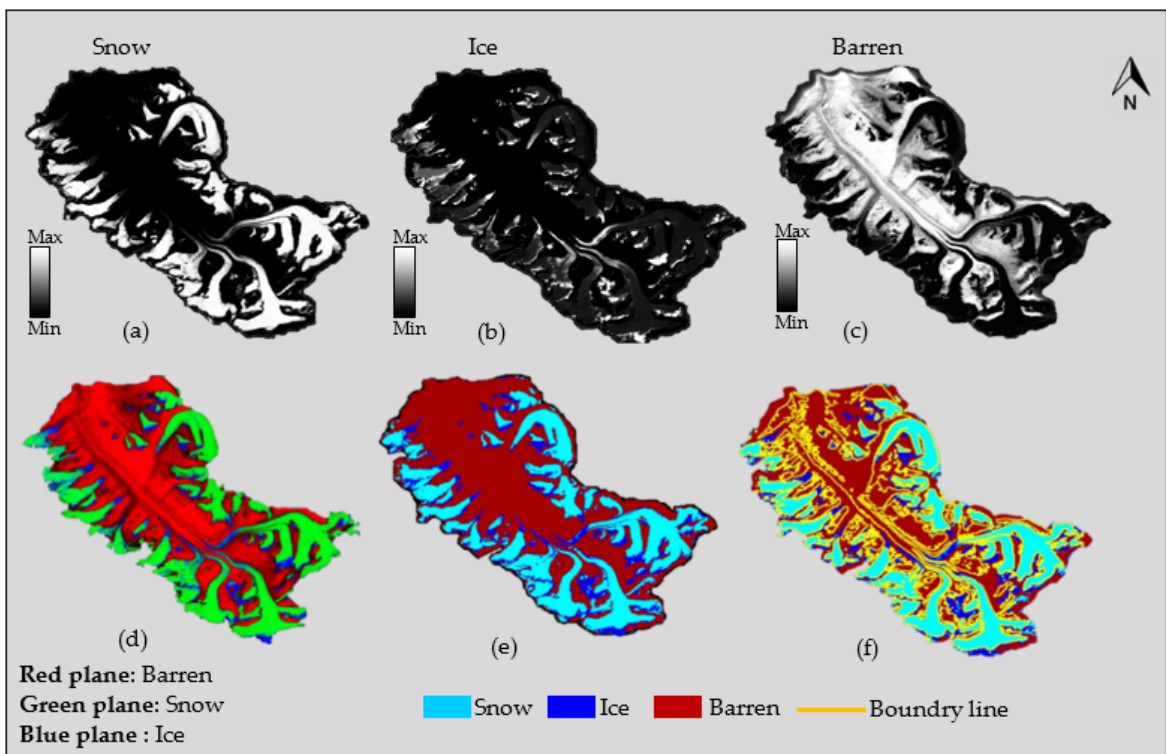

**Figure 6.** ENVINet5-based fractional maps of (**a**) snow, (**b**) ice, and (**c**) barren; (**d**) activation map, (**e**) classified map; and (**f**) boundary extracted classified map.

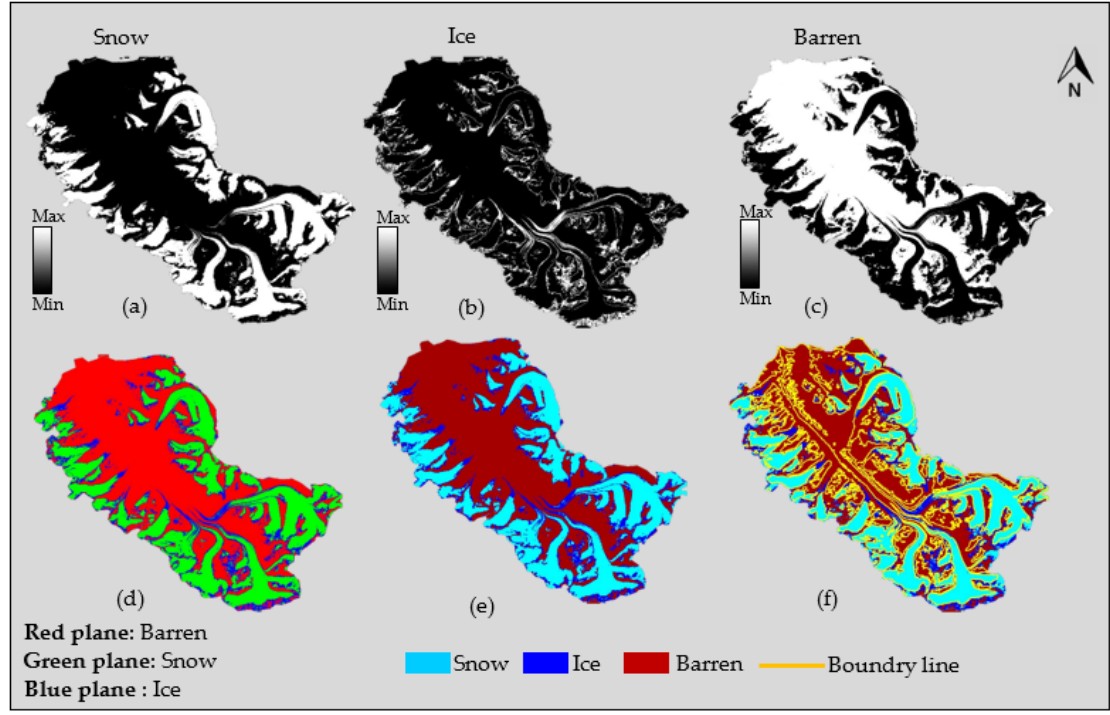

**Figure 7.** ANN-based fractional maps of (**a**) snow, (**b**) ice, and (**c**) barren; (**d**) activation map; (**e**) classified map; and (**f**) boundary extracted classified map.

Moreover, quantitative analysis is also essential to measure the performance of methods, i.e., ENVINet5 and ANN. Table 1 shows the accuracy assessment computed ENVINet5 and ANN classified maps. From experimental outcomes, it has been confirmed that deep

learning plays an effective role in generating classified maps with an OA of 91.89% and k of 0.8778. On the other hand, NN generated classified maps with an OA of 88.38% and k of 0.8241. Moreover, the error rates, i.e., OE and CE, are less (OE, ranging from 7.05–8.01% and CE, ranging from 7.53–8.55%) in deep learning as compared to ANN (OE, ranging from 9.69–14.02% and CE, ranging from 6–17.41%).

**Table 1.** Accuracy assessment of deep learning model and neural network (NN).

| | | Pixel Data (%) | | | Error Rate (%) | | Accuracy | | |
|---|---|---|---|---|---|---|---|---|---|
| | | RT | CT | NC | OE | CE | PA (%) | UA (%) | k |
| ENVI-Net5 | Snow | 37.40 | 37.60 | 34.77 | 7.05 | 7.53 | 92.95 | 92.47 | 0.8797 |
| | Ice | 32.91 | 33.11 | 30.27 | 8.01 | 8.55 | 91.99 | 91.45 | 0.8725 |
| | Barren | 29.69 | 29.30 | 26.86 | 9.54 | 8.33 | 90.46 | 91.67 | 0.8815 |
| | | OA = 91.89; Kc = 0.8778 | | | | | | | |
| ANN | Snow | 39.75 | 39.84 | 40.00 | 11.06 | 11.27 | 88.94 | 88.73 | 0.8129 |
| | Ice | 28.22 | 30.86 | 28.84 | 9.69 | 17.41 | 90.31 | 82.59 | 0.7575 |
| | Barren | 32.03 | 29.30 | 31.16 | 14.02 | 6 | 85.98 | 94 | 0.9117 |
| | | OA = 88.38; Kc = 0.8241 | | | | | | | |

Note: RT: Reference total; CT: column total; NC: number of correct pixels; OE: omission error; CE: commission error; PA: producer's accuracy; UA: user's accuracy; Kc: kappa coefficient.

However, the error rate in ENVINet5 is still high due to various reasons such as the spatial resolution of input data, parameter selection during the training process, or sample selection in accuracy assessment procedures. Still, these results can be improved by implementing more optimized models in deep learning data classification, quantification of uncertainties, and efficient unsupervised deep learning models. Deep learning using the TensorFlow library offers various advantages such as generating classified maps and shapefiles using an activation raster, suitable to extract the one class or multiclass features, and easy to train the algorithm without explicitly programming. Authors of [62] demonstrated the application of ENVINet5 in the estimation of visible land boundaries from unmanned aerial vehicle (UAV) imagery. It was concluded that the ENVINet5 offered the more flexible georeferenced boundary mapping approach, which does not require any explicitly programming as compared to the CNN model.

Different versions of U-Net were also explored in the continuous and accurate monitoring of glaciers. The authors of [63] proved the applicability of channel-attention U-Net with conditional random field (CRF) in the identification and extraction of glaciers in the Pamir Plateau. It allows for a reduction of background noise but faces the problem of misclassification when distinguishing the other geological feature with high spectral similarity (debris-covered glaciers). The clouds and shadows are the major issues involved in underestimating the accuracy of the deep learning algorithm. Therefore, future studies may involve the incorporation or fusion of the optical dataset and microwave dataset (i.e., SAR or Scatterometer), which is less affected by the clouds, shadow, and other atmospheric effects. It will allow for the extraction of richer features and overcome the problem of underestimated debris-covered glaciers. Some of the other deep learning models, i.e., DeepLabv3+ [64], and GlacierNet's CNN [65] performed well in debris-covered glacier (DCG) boundary detection as compared to U-Net. Therefore, these models may also be tested over the Himalayan glaciers.

In our study, ENVINet5 performed well enough using cloud-free Landsat-8 over the Bara Shigri glacier to be utilized for monitoring and mapping the glacier boundary. However, the results may be affected by clouds or topographic effects such as a shadow, which is very common in the rugged terrain mountainous region. Therefore, the impact of topographic corrections could also be tested on the ENVINet5 or improved models. Deep learning is one of the most promising tools in data classification as compared to machine learning methods. Other important applications may include the mapping of the glacial

lake, rock glaciers, and snow-covered accumulation zone [65]. Moreover, some hydrological parameters, i.e., temperature, humidity, groundwater level, and rain, need to be introduced in the deep learning model to incorporate physical equations into neural network formulations [66]. From geomorphological and hydrological perspectives, the identification of critical nodes on river networks along with geomorphic and climatic properties also needs to be explored by incorporating deep learning [67,68]. Future recommendations involve the development of unsupervised deep learning algorithms, interferometric data processing, addressing large-scale nonlinear optimization problems, and quantification of uncertainties.

## 5. Conclusions

In this study, the potential of ENVINet5 (U-Net)-based deep learning was demonstrated in the delineation of glacier boundaries over the Bara Shigri glacier in the western Himalayas using the Landsat-8 dataset. From the experimental outcomes, it is apparent that ENVINet5-based deep learning has the potential to improve the efficacy of glacier classification and boundaries mapping. The ENVINet5 allows more flexible georeferenced boundary mapping and an easy-to-train the algorithm without explicit programming. However, the error rate in ENVINet5 needs to be improved through the fine-tuning of the extracted results to extract more accurate glacier regions. In future work, the hydrological parameters, i.e., temperature, humidity, groundwater level, and precipitation, could be incorporated into the deep learning model to perform the time series analysis of glacier movement in the Himalayas. Moreover, sophisticated models could also be integrated to increase the potential applications of deep learning in cryosphere studies. It is expected that the results of the present work will provide practical guidance to researchers and scientists regarding the effective utilization of deep learning in various remote sensing applications and overcome the existing challenges.

**Author Contributions:** V.S.: Conceptualization methodology, investigation, original draft preparation, and validations; R.K.T.: algorithm formulation, supervision, article preparation, and visualization; S.S. and R.K.: cross-validation, testing, and evaluation; B.R.P.: reviewing, editing, and revision. All authors have read and agreed to the published version of the manuscript.

**Funding:** This research work is financially supported by Women Scientist Scheme-A (WOS-A) Project (Grant no. SR/WOS-A/ET-55/2019) by the Department of Science and Technology (DST), Govt. of India and Teachers Associateship for Research Excellence (TARE) Project (Grant no. TAR/2019/000354) by Science and Engineering Research Board (SERB), Govt. of India.

**Institutional Review Board Statement:** Not applicable.

**Informed Consent Statement:** Not applicable.

**Data Availability Statement:** Landsat-8 satellite data and STRM-based DEM can be downloaded free from the USGS web portal (https://earthexplorer.usgs.gov/) (accessed on 4 May 2021). Pléiades constellation (Pléiades–HR-1A and Pléiades–HR-1B) satellite data are available at the Google Earth Engine (GEE) website (https://earth.google.com/) (accessed on 4 May 2021).

**Acknowledgments:** The authors would like to thank the Land Processes Distributed Active Archive Center (LP-DAAC)/National Snow and Ice Data Center (NSIDC), United States Geological Survey, National Aeronautics and Space Administration (NASA), and Centre National d'Etudes Spatiales (CNES) for providing the Landsat-8 and Pléiades data, respectively. Thanks are due to the anonymous reviewers/Editor for their constructive comments and suggestion to improve the earlier version of the manuscript.

**Conflicts of Interest:** We wish to confirm that there are no known conflict of interest associated with this publication, and there has been no significant financial support for this work that could have influenced its outcome.

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
