# Peer review of "Glacier Boundary Mapping Using Deep Learning Classification over Bara Shigri Glacier in Western Himalayas"

_sustainability, doi:10.3390/su142013485_

Round 1
Reviewer 1 Report
Although the manuscript has the potential to advance our understanding on Delineation of Glacier Boundaries, the specific implication of this study need to address. Please address the comments to improve the quality of your article.
- I believe this study could be extremely beneficial in the context of hydrology, geomorphology, and stream network features on the landscape. Please explain implication in details in a separate section. I recommend that the authors provide the following references: (a) Sarker et al. (2019), Critical Nodes in River Networks, Scientific Reports. https://www.nature.com/articles/s41598-019-47292-4, (b) Gao et al. (2022), Analyzing the critical locations in response of constructed and planned dams on the Mekong River Basin for environmental integrity, Environmental Research Communications, https://iopscience.iop.org/article/10.1088/2515-7620/ac9459.
Author Response
Reviewer-1 Comments:
|
Q1 |
I believe this study could be extremely beneficial in the context of hydrology, geomorphology, and stream network features on the landscape. Please explain implication in details in a separate section. I recommend that the authors provide the following references: (a) Sarker et al. (2019), Critical Nodes in River Networks, Scientific Reports. https://www.nature.com/articles/s41598-019-47292-4, (b) Gao et al. (2022), Analyzing the critical locations in response of constructed and planned dams on the Mekong River Basin for environmental integrity, Environmental Research Communications, https://iopscience.iop.org/article/10.1088/2515-7620/ac9459. |
|
Ans: |
We appreciate the reviewers to highlight other important perspective of the hydrological analysis. We have added few more details in the future studies with respect to hydrological analysis as follows: Page 9 “As in our study, ENVINet5 performed well enough using cloud-free Landsat-8 over the Bara Shigri glacier to be utilized for monitoring and mapping the glacier boundary. However, the results may be affected by clouds or topographic effects like a shadow which is very common in the rugged terrain mountainous region. Therefore, the impact of topographic corrections could also be tested on the ENVINet5 or improved models. Deep learning is one of the most promising tools in data classification as compared to machine learning methods. Other important applications may include the mapping of the glacial lake, rock glaciers, and snow-covered accumulation zone [66]. Moreover, some hydrological parameters i.e., temperature, humidity, groundwater level and rain need to be introduced in the deep learning model to incorporate physical equations into neural network formulations [67]. From geomorphological and hydrological perspectives, the identification of critical nodes on river networks along with geomorphic and climatic properties also needs to be explored by incorporating deep learning [68,69]. Future recommendations involve the development of unsupervised deep learning algorithms, interferometric data processing, addressing large-scale nonlinear optimization problems, and quantification of uncertainties.”
Same References Added under Reference Section. |
|
|
*** |
We greatly appreciate the constructive comments and valuable suggestions from the anonymous reviewers/Editor to improve the earlier version of the manuscript and also, enhance the scope of the paper.
If there are any further modifications are required in the manuscript, we will please do that.
Thank you for reviewing our manuscript and helping us to significantly improve the earlier version of the manuscript in a better way.
Best regards
Corresponding Author

Reviewer 2 Report
The above manuscript is in the field of environmental engineering and is a very applicable, essential and useful research in this field. The novelty is good but I think some minor modifications are needed as bellow:
1- The English is good but it should be modified in rare cases grammatically
2- The abstract is very direct and does not show the importance and the unique solution of the study
3-The introduction part needs more new research to be mentioned
4-The major issue of the manuscript is the number of input variables for hydrological analysis. I think some new parameters like humidity or maybe the groundwater level or subsurface flow or even the discharge of the tributaries are important. Otherwise a deep learning does not make sense. Authors should clarify this in the text
5-The conclusion part needs to be rewritten one more time to show the strong points of the study
Kind regards
Author Response
Reviewer-2 Comments:
|
Q1 |
The English is good but it should be modified in rare cases grammatically |
|
Ans: |
As per suggestion from the reviewer, we have cross-checked the grammatical errors. |
|
|
*** |
|
Q2 |
The abstract is very direct and does not show the importance and the unique solution of the study |
|
Ans: |
As per suggestion from the reviewer, we have revised the abstract section as follows: Page 1 “Abstract: Glacier, snow, and ice are the essential components of the Himalayan cryosphere and provide a sustainable water source for different applications. Continuous and accurate monitoring of glaciers allows the forecasting analysis of natural hazards and water resource management. In past literature, different methodologies such as spectral unmixing, object-based detection, and a combination of various spectral indices are commonly utilized for mapping snow, ice and glaciers. Most of these methods require human intervention in feature extraction, training of the models, and validation procedures which may create bias in the implementation approaches. In this study, the deep learning classifier based on ENVINet5 (U-Net) architecture has been demonstrated in the delineation of glacier boundaries along with snow/ice over the Bara Shigri glacier (Western Himalayas), Himachal Pradesh, India. Glacier monitoring with Landsat data takes the advantage of a long coverage period and finer spectral/spatial resolution with wide coverage on a larger scale. Moreover, deep learning utilizes the semantic segmentation network to extract glacier boundaries. Experimental outcomes confirm the effectiveness of deep learning (overall accuracy, 91.89% and Cohen’s kappa coefficient, 0.8778) compared to the existing artificial neural network (ANN) model (overall accuracy, 88.38% and Kappa coefficient, 0.8241) in generating accurate classified maps. This study is vital in the study of the cryosphere, hydrology, agriculture, climatology, and land-use/ land-cover analysis.” |
|
|
|
|
Q3 |
The introduction part needs more new research to be mentioned |
|
Ans: |
As per suggestion from the reviewer, we have added the few important details under “Introduction section” as follows: Page 2-3 “With the availably of high-resolution satellite datasets and computations power, deep learning found numerous applications in earth observations through remote sensing such as data fusion, object detection, target detection and scene classifications [30]. Recently, some attempts have been made to deep learning on glacier studies such as simulation and reconstruction of annual glacier-wide surface mass balance (SMB) using deep learning-based artificial neural network (ANN) model [31], autonomous estimation of rock glaciers using deep learning based CNN model and object-based image analysis [32], debris-covered glacier (DCG) mapping using deep learning based CNN model [33], Alpine glacier mapping using deep learning based CNN model [34] and many more. Deep ANN as an alternative to the conventional model is the nonlinear model that solves the regression problems for predicting the various glacier components such as ice thickness [35] and mass balance [36]. The deep CNN model requires larger sample datasets to train the models and helps in the recognition of repeated patterns in different applications such as snow avalanches [37–40], permafrost thaw slumps [41], and sea-ice mapping [42]. However, very few studies have focused on glacier boundary mapping using deep learning models which may be beneficial in climate change studies. Furthermore, deep learning is yet to be explored for different applications with different remote sensing datasets because the complexity increases with: (a) high spectral/spatial resolution to extract detailed information; (b) a large number of training samples to improve the recognition; and (c) depth of the deep learning model to learn complex distribution [29].” |
|
|
|
|
Q4 |
The major issue of the manuscript is the number of input variables for hydrological analysis. I think some new parameters like humidity or maybe the groundwater level or subsurface flow or even the discharge of the tributaries are important. Otherwise, a deep learning does not make sense. Authors should clarify this in the text |
|
Ans: |
We appreciate the reviewers to highlight other important perspective of the hydrological analysis. Currently, the main focus of this study is to delineate the glacier boundaries along with snow and ice coverage using a deep learning classifier over a part of the western Himalayas because very few studies have focused on glacier boundary mapping using deep learning models which may be beneficial in climate change studies. However, we have added few more details in the future studies with respect to hydrological analysis as follows: Page 9 “As in our study, ENVINet5 performed well enough using cloud-free Landsat-8 over the Bara Shigri glacier to be utilized for monitoring and mapping the glacier boundary. However, the results may be affected by clouds or topographic effects like a shadow which is very common in the rugged terrain mountainous region. Therefore, the impact of topographic corrections could also be tested on the ENVINet5 or improved models. Deep learning is one of the most promising tools in data classification as compared to machine learning methods. Other important applications may include the mapping of the glacial lake, rock glaciers, and snow-covered accumulation zone [66]. Moreover, some hydrological parameters i.e., temperature, humidity, groundwater level and rain need to be introduced in the deep learning model to incorporate physical equations into neural network formulations [67]. From geomorphological and hydrological perspectives, the identification of critical nodes on river networks along with geomorphic and climatic properties also needs to be explored by incorporating deep learning [68,69]. Future recommendations involve the development of unsupervised deep learning algorithms, interferometric data processing, addressing large-scale nonlinear optimization problems, and quantification of uncertainties.” |
|
|
|
|
Q5 |
The conclusion part needs to be rewritten one more time to show the strong points of the study. |
|
Ans: |
As per suggestion from the reviewer, we have revised the conclusion section as follows: Page 10 “In this study, the potential of ENVINet5 (U-Net) based deep learning has been demonstrated in the delineation of glacier boundaries over the Bara Shigri glacier in Western Himalayas using the Landsat-8 dataset. From the experimental outcomes, it is apparent that ENVINet5-based deep learning has the potential to improve the efficacy of glacier classification and boundaries mapping. The ENVINet5 allows more flexible georeferenced boundary mapping and easy to train the algorithm without explicitly programming. However, the error rate in ENVINet5 needs to be improved through the fine-tuning of the extracted results to extract more accurate glacier regions. In future work, the hydrological parameters i.e., temperature, humidity, groundwater level and precipitation could be incorporated into the deep learning model to perform the time series analysis of glacier movement in the Himalayas. Moreover, sophisticated models could also be integrated to increase the potential applications of deep learning in cryosphere studies. It is expected that the results of the present work will provide practical guidance to researchers and scientists regarding the effective utilization of deep learning in various remote sensing applications and overcome the existing challenges.” |
|
|
|
We greatly appreciate the constructive comments and valuable suggestions from the anonymous reviewers/Editor to improve the earlier version of the manuscript and also, enhance the scope of the paper.
If there are any further modifications are required in the manuscript, we will please do that.
Thank you for reviewing our manuscript and helping us to significantly improve the earlier version of the manuscript in a better way.
Best regards
Corresponding Author

Reviewer 3 Report
Delineation of Glacier Boundaries with Deep Learning Classification over Bara Shigri Glacier in Western Himalayas
Interesting paper; however, the title is not clear enough. I checked the plagiarism we are turning and found out that the scores are less than 20% therefore this particular manuscript can be moved revealing level.
Research gap is not clearly presented therefore it is advisable to state research gap.
Abstract: Glacier, snow, and ice are the essential parameters of the Himalayan cryosphere and pro- 15 vide a sustainable water source for different applications.
What do you mean by parameters? Something measurable?
Acceptable introduction. a good number of references was cited in introduction section. However, the research gap can be most strengthened.
Figure 3 has to be explained in detail.
Accuracy assessment has to be explained in detail. This is an important section.
If you look closely look at the first paragraph of the results and discussion section, it is more or less methodology. So you need to understand what should be presented in results and discussion. It is just results and discussion; whatever comes under the methodology should not be presented in the results and discussion. Therefore, you need to restructure the manuscript.
"In this study, we have implemented ENVINet5 (U-Net) based deep learning to estimate the various cryosphere parameters"
This is not clear. What do you mean by parameters? Parameters should be something measurable
Why didn't you use other landsat images I understand you have used landsat8 images.
Author Response
Reviewer-3 Comments:
|
Q1 |
Delineation of Glacier Boundaries with Deep Learning Classification over Bara Shigri Glacier in Western Himalayas. Interesting paper; however, the title is not clear enough. |
|
Ans: |
As per suggestion from the reviewer, we have revised the Title of the article as follows: Page 1 “Delineation of Glacier Boundary Mapping with Deep Learning Classification over Bara Shigri Glacier in Western Himalayas” |
|
|
|
|
Q2 |
Research gap is not clearly presented therefore it is advisable to state research gap. |
|
Ans: |
As per suggestion from the reviewer, we have added the few details to highlight the research gap as follows: Page 3 “With the availably of high-resolution satellite datasets and computations power, deep learning found numerous applications in earth observations through remote sensing such as data fusion, object detection, target detection and scene classifications [30]. Recently, some attempts have been made to deep learning on glacier studies such as simulation and reconstruction of annual glacier-wide surface mass balance (SMB) using deep learning-based artificial neural network (ANN) model [31], autonomous estimation of rock glaciers using deep learning based CNN model and object-based image analysis [32], debris-covered glacier (DCG) mapping using deep learning based CNN model [33], Alpine glacier mapping using deep learning based CNN model [34] and many more. Deep ANN as an alternative to the conventional model is the nonlinear model that solves the regression problems for predicting the various glacier components such as ice thickness [35] and mass balance [36]. The deep CNN model requires larger sample datasets to train the models and helps in the recognition of repeated patterns in different applications such as snow avalanches [37–40], permafrost thaw slumps [41], and sea-ice mapping [42]. However, very few studies have focused on glacier boundary mapping using deep learning models which may be beneficial in climate change studies. Furthermore, deep learning is yet to be explored for different applications with different remote sensing datasets because the complexity increases with: (a) high spectral/spatial resolution to extract detailed information; (b) a large number of training samples to improve the recognition; and (c) depth of the deep learning model to learn complex distribution [29].” |
|
|
|
|
Q3 |
Abstract: Glacier, snow, and ice are the essential parameters of the Himalayan cryosphere and provide a sustainable water source for different applications. What do you mean by parameters? Something measurable? |
|
Ans: |
· To avoid any confusion, we have now revised the term “Components”. · The objectives of the study are framed as (a) computation of snow/ice maps using ENVINet5 (based on U-Net) deep learning, (b) extraction of glacier boundary, (c) comparative analysis with the existing ANN-based model, and (d) validation and accuracy assessment procedures for the classified outcomes. |
|
Q4 |
Acceptable introduction. A good number of references was cited in introduction section. However, the research gap can be most strengthened. |
|
Ans: |
As per suggestion from the reviewer, we have added the few details to highlight the research gap as follows: Page 3 “With the availably of high-resolution satellite datasets and computations power, deep learning found numerous applications in earth observations through remote sensing such as data fusion, object detection, target detection and scene classifications [30]. Recently, some attempts have been made to deep learning on glacier studies such as simulation and reconstruction of annual glacier-wide surface mass balance (SMB) using deep learning-based artificial neural network (ANN) model [31], autonomous estimation of rock glaciers using deep learning based CNN model and object-based image analysis [32], debris-covered glacier (DCG) mapping using deep learning based CNN model [33], Alpine glacier mapping using deep learning based CNN model [34] and many more. Deep ANN as an alternative to the conventional model is the nonlinear model that solves the regression problems for predicting the various glacier components such as ice thickness [35] and mass balance [36]. The deep CNN model requires larger sample datasets to train the models and helps in the recognition of repeated patterns in different applications such as snow avalanches [37–40], permafrost thaw slumps [41], and sea-ice mapping [42]. However, very few studies have focused on glacier boundary mapping using deep learning models which may be beneficial in climate change studies. Furthermore, deep learning is yet to be explored for different applications with different remote sensing datasets because the complexity increases with: (a) high spectral/spatial resolution to extract detailed information; (b) a large number of training samples to improve the recognition; and (c) depth of the deep learning model to learn complex distribution [29].” |
|
|
|
|
Q5 |
Figure 3 has to be explained in detail. |
|
Ans: |
As per suggestion from the reviewer, we have added the few details to highlight the research gap as follows: Page 3 “This study has been performed over a Bara Shigri glacier, located near the Chandra River basin, nearby Chandra valley on the northern ridge of Pir-Panjal range, Lahaul –Spiti, Himachal Pradesh, India using Landsat-8 satellite dataset as shown in Fig. 2. This glacier is the second longest glacier (28 km) in the Himalayas, after Gangotri and situated between geographical coordinates 32°05’ N - 32°17’ N latitude and 77°33’ N -77°48’ E longitude. It covers the 131 km2 area with glacierized area ranging between 3920 m (13,071 ft) – 6550 m (20,876 ft) a.m.s.l. [43,44]. As the largest glacier in Himachal Pradesh (Indian State), it provides valuable information about climate change studies [45]. Various studies also explored the retreating of the Bara Shigri glacier with an average terminus retreat of 1100.2±32.1 m (22.5±0.7 m per year) from 1965 to 2014 [46,47]. The satellite dataset was acquired on a cloud-free clear daytime (20-Sept-2017) from Landsat‒8 OLI (Operational Land Imager) with 30 m spatial resolution. The Shuttle Radar Topography Mission (SRTM) based Digital Elevation Model (DEM, version 4) with 30 m spatial resolution data was utilized to analyze the topography of the Bara Shigri glacier as shown in Fig. 3. Fig. 3(b) and 3(c) represent the aspect (compass direction/azimuth i.e., terrain surface faces, generally measured usually measured in degrees from north) and slope (steepness of terrain i.e., degree of inclination of land surface to the horizontal plane) respectively generated from STRM-DEM. This glacier has heterogenic surface characteristics i.e., clean ice (located in the accumulation zone) and debris cover (located in the lower ablation zone).” |
|
|
|
|
Q6 |
Accuracy assessment has to be explained in detail. This is an important section. |
|
Ans: |
As per suggestion from the reviewer, we have added the details of parameters used in the manuscript as follows: Page 7 “To test the efficacy of ENVINet5-based deep learning, an accuracy assessment is computed for the classified dataset. The various important parameters of accuracy assessment involve the producer’s accuracy (PA) i.e., probability of correct classification of reference pixels; user’s accuracy (UA) i.e., probability of pixels falling under precise class category; commission error (CE) i.e., classified sites probability for incorrect classification; omission error (OE) i.e., the probability to classify reference pixels correctly; overall accuracy (OA) i.e., collective accuracy map for all the classes and Cohen’s kappa coefficient (k) i.e., the distinction between actual and expected agreement and computed as follow: Where represents the total count of observations; shows the errors matrix count of rows and columns; depicts the row and column observation belonging to class ; represents the true values of class whereas shows the predicted values of class . For sample selection, stratified random sampling is utilized to select the samples from a population, which can be partitioned into subpopulations. The outcome of the ENVINet5-based deep learning is compared with the conventional ANN model i.e., feed-forward NN.” |
|
|
|
|
Q7 |
If you look closely look at the first paragraph of the results and discussion section, it is more or less methodology. So, you need to understand what should be presented in results and discussion. It is just results and discussion; whatever comes under the methodology should not be presented in the results and discussion. Therefore, you need to restructure the manuscript. |
|
Ans: |
As per the suggestions from the reviewer, we have revised the “Result and discussion” Section as follows: Page 7-8 The ENVI v5.6 deep learning module v1.1 generates the activation rasters in the form of fractional maps or grayscale (value lies between ‘0’ to ‘1’) of each class category, i.e., snow, ice, and barren, as shown in Fig. 5(a)-(c). The value ‘0’ means the minimum existence of the class category and the value ‘1’ means the maximum existence of the class category. These fractional maps allow the accurate detection of particular class categories. These fractional maps can be visualized in combined form by setting the ice in the blue plane, snow in the green plane, and barren in the red plane, as shown in Figure 5(d). Furthermore, the class membership of each pixel can be measured by implementing the threshold algorithm. Here, we have used Otsu as an automatic threshold search method to generate the classified image from the activation rasters, as shown in Figure 5(e). If the results are not satisfactory, the user can reset the threshold value or threshold algorithm (in the ENVI v5.6 deep learning module v1.1) to regenerate the accurate classified image from the activation raster instead of recomputing the entire deep learning classification process. Once the model is trained, it can also be used for different multitemporal images of the same satellite dataset with the same number of spectral bands only. In addition, the ENVINet5 also allows the creation of the polygon shapefile from the class activation raster, to draw and estimate the boundaries around the different features as shown in Figure 5(f). ENVINet5 delivers a robust method for learning complex spatial and spectral information which may help in extracting the essential features from a complex dataset, regardless of their different attributes e.g., shape, color, size, and barren. For the cross-referencing, the outcomes of ENVINet5 have been compared with the conventional ANN model i.e., feed-forward NN as shown in Fig. 6. Fig. 6(a)-(c) represent the ANN activation maps (i.e., snow, ice, and barren) computed generated via ANN model. Fig. 6 (d)-(f) represent the combined form (Red plane: barren; Green plane: snow; and blue plane: ice) of fractional maps, classified maps and boundary maps. Moreover, quantitative analysis is also essential to measure the performance of methods i.e., ENVINet5 and ANN. Table 1 shows the accuracy assessment computed ENVINet5 and ANN classified maps. From experimental outcomes, it has been confirmed that deep learning plays an effective role in generating classified maps with an OA of 91.89% and k of 0.8778. On the other hand, NN generated classified maps with an OA of 88.38% and k of 0.8241. Moreover, the error rates, i.e., OE and CE, are less (OE, ranging from 7.05 – 8.01% and CE, ranging from 7.53 – 8.55%) in deep learning as compared to ANN (OE, ranging from 9.69 – 14.02% and CE, ranging from 6 – 17.41%).
We have moved the certain part from the “Result and Discussion” Section to “Methodology” Section as follows: Page 5-6 “Classification allows the quantitative analysis of remote sensing image data to categorise all the pixels in imagery into a specific set of class categories. In previous literature, many machine learning or deep learning classifiers were developed and summarized by various authors in different applications with different remote sensing datasets [53,54]. Among the various classifiers, deep learning approaches attract the interest of many researchers due to their potential for efficient extraction of discriminative features and their representation in the real domain [55]. Some of the prominent deep learning-based classifiers are CNN to learn the learning of low- and high-level features [55]; RNN to solve the problem of gradient diminishing [56]; and DRL to learn the best actions for given states using a feedback–reward system [57]; GNN to learning from unstructured data [58], SAE to automatically learn deep features [59]; and GAN that contains the generator and a discriminator to learn a latent space [60]. Here, we have implemented the deep learning classifier based on ENVINet5 and ENVI Net-Multi. The deep learning version 1.1 is an add-on module in ENVI (Environment for Visualizing Images) image processing software version 5.6 (2020) image analysis. Fig. 5 represents the ENVINet5 architecture: (a) input patch (grey color), (b) feature map (orange color); (c) 3 × 3 convolution (green color); (d) feature fusion (yellow color); (e) dimensionality reduction by maximum pooling (blue color); (f) co-convolution (red color); and (g) 1 × 1 convolution (purple color). This architecture is mask-based, encoder-decoder architecture to categorise the pixel data in imagery. In ENVI deep learning model, the ENVINet5 and ENVI Net-Multi are specially designed for a single class and multiple classes, respectively. These architectures are based on the U-Net model (a channel-attention U-Net) with five levels with twenty-three convolutional layers [61]. The robustness of the model towards the distortion can be controlled by down-sampling. The restoration and decoding of the abstract features can be done by up-sampling. Here, ENVINet-Multi has been utilized to classify the multiple class categories. In this model, the multiple features, i.e., snow, ice, and barren (including the mixture of barren or trees), are extracted based on input dataset spectral and spatial properties with the knowledge of field data. To train the TensorFlow, multiple on-screen training samples (50 - 60 polygons) from each class category with respect to the knowledge of ground truth information. The various training parameters for ENVINet-Multi have been set as (a) number of epochs: 25; (b) patch sampling rate: 16; (c) class weight: 2.5; (d) loss weight: 0.5; (e) number of patches per epoch: 200; (f) patch size: 464×464 pixels; and (g) number of patches per batch: 2. In the training process, it repeatedly exposes label rasters to a model and thus, converts the spectral and spatial information of the raster into the thematic map with the help of prepared training samples. The data is split into two parts, i.e., training (80%) and validation (20%) [62]. At last, the TensorFlow mask classification tool is utilized to classify the input dataset with the help of a trained model. This entire process may require a significant amount of time as per the computation availability. While implementing the ENVINet5-based deep learning on the Landsat-8 dataset, the computer specification includes the Intel Xeon 3.2 2400 MHz 8.25 4C CPU, 16 GB of RAM, 512 GB of SSD, and NVIDIA Quadro P620 2 GB (4) MDP GFX.” |
|
|
|
|
Q8 |
"In this study, we have implemented ENVINet5 (U-Net) based deep learning to estimate the various cryosphere parameters" This is not clear. What do you mean by parameters? Parameters should be something measurable. |
|
Ans: |
· To avoid any confusion, we have now revised the term “Components”. |
|
|
|
|
Q9 |
Why didn't you use other Landsat images I understand you have used landsat8 images. |
|
Ans: |
From the future & climate perspective, we may analyse the multi-temporal changes (annual variation) and Landsat series may be proven as significant. This is the main reason behind the selection of Landsat data. |
|
|
|
We greatly appreciate the constructive comments and valuable suggestions from the anonymous reviewers/Editor to improve the earlier version of the manuscript and also, enhance the scope of the paper.
If there are any further modifications are required in the manuscript, we will please do that.
Thank you for reviewing our manuscript and helping us to significantly improve the earlier version of the manuscript in a better way.
Best regards
Corresponding Author

Round 2
Reviewer 1 Report
Thanks.
Reviewer 3 Report
I am happy for the revisions.